Transcriptome association studies of neuropsychiatric traits in African Americans implicate PRMT7 in schizophrenia

Fiorica Peter N. 1 2
http://orcid.org/0000-0003-1365-9667 Wheeler Heather E. 2 3 4 5 hwheeler1@luc.edu
1 Department of Chemistry and Biochemistry, Loyola University Chicago , Chicago, IL , USA
2 Department of Biology, Loyola University Chicago , Chicago, IL , USA
3 Program in Bioinformatics, Loyola University Chicago , Chicago, IL , USA
4 Department of Computer Science, Loyola University Chicago , Chicago, IL , USA
5 Department of Public Health Sciences, Loyola University Chicago , Maywood, IL , USA
Khiabanian Hossein
Electronic publication date: 2019 Sep 26
Publication date: 2019
Volume: 7
Electronic Location ID: e7778
Received 2019 May 13; Accepted 2019 Aug 27
Copyright: © 2019 Fiorica and Wheeler
Copyright year: 2019
Copyright holder: Fiorica and Wheeler
License: This is an open access article distributed under the terms of the Creative Commons Attribution License, which permits unrestricted use, distribution, reproduction and adaptation in any medium and for any purpose provided that it is properly attributed. For attribution, the original author(s), title, publication source (PeerJ) and either DOI or URL of the article must be cited.
License URL: https://creativecommons.org/licenses/by/4.0/

Keywords: GWAS, PrediXcan, Population genetics, Gene expression, Schizophrenia, Bipolar disorder

Funding: National Institutes of Health National Human Genome Research Institute Academic Research Enhancement Award R15 HG009569 Loyola University Chicago Carbon Undergraduate Research Fellowship Loyola University Chicago Mulcahy Scholarship This work was supported by the National Institutes of Health National Human Genome Research Institute Academic Research Enhancement Award R15 HG009569 (PI: Heather E. Wheeler), the Loyola University Chicago Carbon Undergraduate Research Fellowship (Peter N. Fiorica), and the Loyola University Chicago Mulcahy Scholarship (Peter N. Fiorica). The funders had no role in study design, data collection and analysis, decision to publish, or preparation of the manuscript.

==============================
In the past 15 years, genome-wide association studies (GWAS) have provided novel insight into the genetic architecture of various complex traits; however, this insight has been primarily focused on populations of European descent. This emphasis on European populations has led to individuals of recent African descent being grossly underrepresented in the study of genetics. With African Americans making up less than 2% of participants in neuropsychiatric GWAS, this discrepancy is magnified in diseases such as schizophrenia and bipolar disorder. In this study, we performed GWAS and the gene-based association method PrediXcan for schizophrenia (n = 2,256) and bipolar disorder (n = 1,019) in African American cohorts. In our PrediXcan analyses, we identified PRMT7 (P = 5.5 × 10−6, local false sign rate = 0.12) as significantly associated with schizophrenia following an adaptive shrinkage multiple testing adjustment. This association with schizophrenia was confirmed in the much larger, predominantly European, Psychiatric Genomics Consortium. In addition to the PRMT7 association with schizophrenia, we identified rs10168049 (P = 1.0 × 10−6) as a potential candidate locus for bipolar disorder with highly divergent allele frequencies across populations, highlighting the need for diversity in genetic studies.

Introduction

Individuals of recent African ancestry have been grossly underrepresented in genomic studies. African American participants make up about 2.0% of all genome-wide association studies (GWAS) subjects (Sirugo, Williams & Tishkoff, 2019). Specifically, individuals of African ancestry make up only 1.2% of all neuropsychiatric GWAS (Quansah & McGregor, 2018). With the advent of polygenic risk scores, accuracy in disease prediction is critical to the development of precision medicine (Khera et al., 2018); however, the lack of representative diversity in the study of genomics has impacted the accuracy of genetic risk prediction across diverse populations. Despite similar incidences of schizophrenia across European and African ancestry populations (De Candia et al., 2013; Whiteford et al., 2013), Africans have been predicted to have significantly less disease risk than their European counterparts using current GWAS summary statistics (Martin et al., 2017). Inaccuracy in predicting disease risk across populations can lead to further disparities in health and treatment of underrepresented populations. To prevent misclassification of genetic risk, further work in the genetics underlying complex traits in African Americans is needed (Manrai et al., 2016). In an attempt to address this discrepancy in genetic risk prediction, we performed a series of genetic association tests for schizophrenia and bipolar disorder in two cohorts of African American individuals (Manolio et al., 2007; Suarez et al., 2006; Smith et al., 2009).

Schizophrenia and bipolar disorder are two heritable neuropsychiatric disorders whose genetic components have been attributed to the cumulative effect of thousands of loci across the genome (Schizophrenia Working Group of the Psychiatric Genomics Consortium, 2014; Li et al., 2017; Ikeda et al., 2017b). Past work shows that the genetic architectures of these two disorders significantly overlap (Bhalala et al., 2018; Allardyce et al., 2018; The International Schizophrenia Consortium, 2009; Stahl et al., 2019). Up to this point, the largest GWAS of schizophrenia and bipolar disorder comprise hundreds of thousands of individuals primarily of European descent (Stahl et al., 2019; Schizophrenia Working Group of the Psychiatric Genomics Consortium, 2014; Li et al., 2017). While studies of neuropsychiatric diseases in European and Asian ancestry populations continue to grow, the scarcity of studies in African American populations persists (Ikeda et al., 2017a).

To date, one of the largest GWAS of schizophrenia in an African American population was completed by the Genetic Association Information Network (GAIN) (Manolio et al., 2007); however, this study found no single nucleotide polymorphisms (SNPs) to be genome-wide significant and offered little insight into the potential function of genes in schizophrenia in African Americans. The GAIN has also performed one of the largest GWAS of bipolar disorder in African Americans (Manolio et al., 2007; Smith et al., 2009). Similar to the findings of the GAIN GWAS of schizophrenia, Smith et al. (2009) found no SNPs significantly associated with bipolar disorder. In addition to a traditional GWAS using a logistic regression, we performed two gene-level association tests: PrediXcan and MultiXcan (Gamazon et al., 2015; Barbeira et al., 2019). PrediXcan offers a series of advantages to SNP-level analyses in detecting genetic association and functionality. First, since tests are being conducted at the gene level, PrediXcan has a lower multiple-testing burden compared to GWAS. Additionally, our understanding of functional pathways are more easily constructed for genes compared to SNPs. By using gene expression as an intermediate phenotype between genetic variation and complex phenotypes, PrediXcan results can help elucidate genetic mechanisms compared to GWAS. We completed our genetic association tests for schizophrenia in a cohort of 2,256 self-identified African American individuals from the genome-wide linkage scan of African American families (Suarez et al., 2006) and the GAIN (Manolio et al., 2007). For our study of bipolar disorder, we performed these association tests in 1,019 African American individuals from the GAIN. Using these data, we identified one gene significantly associated with schizophrenia and tested it for replication in the Psychiatric Genomics Consortium (PGC) GWAS of schizophrenia (Barbeira et al., 2018; Schizophrenia Working Group of the Psychiatric Genomics Consortium, 2014; Psychiatric GWAS Consortium Bipolar Disorder Working Group, 2011).

Methods

Cohorts

The individuals in the cohorts used in this study were all self-identified African Americans. We acquired genotype and phenotype information for these individuals from the National Center for Biotechnology Information database of Genotypes and Phenotypes (dbGaP). The project was confirmed exempt from human subjects federal regulations under exemption number 4 by the Loyola University Chicago Institutional Review Board (project number 2014). Whole Genome genotypes and phenotypic information were acquired from three separate accessions in dbGaP (Table 1). All genotype information for these three accessions were acquired using Affymetrix Genome-Wide Human SNP Array 6.0, covering 934,940 SNPs. In total, our studies included 2,256 and 1,019 individuals for schizophrenia and bipolar disorder, respectively.

Table 1 Cohort characteristics.

Three separate cohorts were integrated into two main cohorts characterized by phenotype. After merging the two schizophrenia cohorts using PLINK, the number of post-QC SNPs became identical.

dbGaP accession number	phs000021.v3.p2	phs000167.v1.p1	phs000017.v3.p1	
Phenotype	Schizophrenia	Schizophrenia	Bipolar disorder	
Total individuals	2,220	120	1,045	
Cases	1,241	15	359	
Controls	979	105	686	
Post-QC individuals	2,256	2,256	1,019	
Pre-QC SNPs	845,814	909,622	867,411	
Post-QC SNPs	742,015	742,015	721,050	
Post-imputation SNPs (r2 > 0.8, MAF > 0.01)	12,780,487	12,780,487	12,799,548	

Case-control criteria for both phenotypes were determined using the DSM-IV (Diagnostic and Statistical Manual of Mental Disorders) as described previously (Smith et al., 2009). Following the update from DSM-IV to DSM-V, 40 individuals previously identified as cases under DSM-IV were removed from the study because they no longer met the case criteria for DSM-V.

Quality control and imputation

Following the download of data from dbGaP, we isolated genotypes and phenotypes of African Americans from those of European Americans or individuals of unidentified ethnicities for each cohort. We merged the PLINK binary files from the two schizophrenia studies. At this point 906,425 SNPs were genotyped in 2,256 individuals in the schizophrenia cohort and 867,411 SNPs and 1,019 individuals in the bipolar disorder cohort. While the schizophrenia cohort included individuals from GAIN and the genome-wide linkage scan of African American families (Suarez et al., 2006; Manolio et al., 2007), throughout the rest of the paper, we will refer to the combined cohort as GAIN. In each cohort, we removed SNPs with genotyping call rates less than 99% and those that significantly deviated from Hardy–Weinberg equilibrium (P < 1× 10−6). We then removed individuals with excess heterozygosity. Individuals greater than three standard deviations from mean heterozygosity were removed from the study. We used EIGENSOFT smartpca (Patterson, Price & Reich, 2006) to generate the first 10 principal components, which were used to confirm self-identified ancestry (Figs. S1 and S2). After this quality control, we had 742,015 SNPs and 2,256 individuals total in the schizophrenia cohort and 721,050 SNPs and 1,019 individuals in the bipolar disorder cohort (Purcell et al., 2007).

From here, the filtered data from each cohort were uploaded to the University of Michigan Imputation Server for genotype imputation (Das et al., 2016). The genotypes for each cohort were imputed using Eagle version 2.3 for phasing and 1000 Genomes Phase 3 version 5 (1000G) as our reference panel (The 1000 Genomes Project Consortium, 2015). After this, we downloaded the imputed data from the Michigan Imputation Server and converted it to PLINK binary format. We then filtered the data by removing SNPs with imputation r2 < 0.8 and minor allele frequency (MAF) <0.01. At this point, we were left with 12,780,487 and 12,799,548 SNPs in the schizophrenia and bipolar disorder cohorts, respectively. We explored Consortium on Asthma among African-ancestry Populations in the Americas (CAAPA) as an alternative reference panel for imputation, but 1000G imputed more SNPs meeting our filters while simultaneously imputing SNPs with MAFs identical to those imputed from CAAPA at our chosen imputation r2 and MAF thresholds (Fig. S3) (Mathias et al., 2016).

Genome-wide association study

Using PLINK, we performed a logistic regression of the phenotype using the first 10 genotypic principal components as covariates to account for population structure. We used a significance threshold of P < 5 × 10−8 to identify significantly associated SNPs. Plots were generated from PLINK results using the web-based tool LocusZoom (Pruim et al., 2011).

PrediXcan

We performed the gene-based association test PrediXcan on both phenotypes, schizophrenia and bipolar disorder, in this study. PrediXcan functions by predicting an individual’s genetically regulated gene expression levels using tissue-dependent prediction models trained using reference transcriptome data (Gamazon et al., 2015). For our experiments we tested each phenotype across 55 prediction models. Forty-eight of these models were trained on 48 tissues in GTEx version 7 (Barbeira et al., 2018). Six of these models were generated from monocyte transcriptomes of individuals in the Multi-Ethnic Study of Atherosclerosis (MESA) cohort. The MESA models, the most diverse set of published predictors to date, were built from genotypes and transcriptomes of self-identified African American, Hispanic, and European individuals (Mogil et al., 2018). These models can be found at http://predictdb.org/. We also used a model built from dorsolateral prefrontal cortex (DLPFC) data from the CommonMind Consortium (Huckins et al., 2019). To impute the gene expression levels, the PLINK binary files from each cohort had to be converted to PrediXcan dosage files. To do this, we used the conversion script provided at https://github.com/hakyimlab/PrediXcan/tree/master/Software. After predicting a genetically regulated level of expression, we tested each expression level for association with the phenotype of interest. Since PrediXcan does not have a flag for performing a logistic regression with covariates, we performed a logistic regression of the phenotype with the first 10 principal components to generate a residual phenotype in order to account for population structure. We then performed a linear regression with the residual phenotype and gene expression level for each gene.

Following the PrediXcan association tests, we adjusted for multiple testing using the adaptive shrinkage approach implemented in the R package ashr (Stephens, 2017). Using this package, we calculated the local false sign rate (lfsr) for each test, which is similar to traditional false discovery rate approaches, but takes into account both the effect sizes and standard errors of each gene-tissue pair (n = 248,605). In addition, this empirical Bayes approach uses the assumption that the distribution of actual effects is unimodal with the mode at 0. We set our significance threshold for gene-tissue pairs at lfsr < 0.2.

Due to the dearth of African American neuropsychiatric cohorts, replication could not be completed in an independent African American cohort. To validate our results, we compared our findings to the association results of a meta-analysis of the PGC GWAS summary statistics completed using S-PrediXcan (Barbeira et al., 2018; Schizophrenia Working Group of the Psychiatric Genomics Consortium, 2014; Psychiatric GWAS Consortium Bipolar Disorder Working Group, 2011). We performed S-PrediXcan in both phenotypes using predictors from GTEx. The Schizophrenia Working Group of the Psychiatric Genomics Consortium (2014) study originally included 36,989 cases and 113,075 controls composed of about 96.5% European and 3.5% Asian individuals. The Psychiatric GWAS Consortium Bipolar Disorder Working Group (2011) study included 7,481 cases and 9,250 controls of primarily European and Asian descent.

MultiXcan

Following imputation of gene expression levels, we performed MultiXcan (Barbeira et al., 2019), a gene-based association test that combines information across multiple tissues while taking their correlation into account. Using the predicted expression levels in 48 tissues across GTEx, we performed MultiXcan on both of the disease phenotypes.

Results

Schizophrenia gene-based association study

To better understand the genetic architecture of schizophrenia in African Americans, we performed transcriptome-wide association studies using prediction models built in 55 tissues. In the GAIN cohort of 2,256 individuals (969 controls and 1,287 cases), we predicted gene expression across 48 tissues in GTEx, six models built from monocytes across MESA, and DLPFC from CommonMind (Barbeira et al., 2018; Wheeler et al., 2016; Mogil et al., 2018; Huckins et al., 2019).

PRMT7 was the most significantly associated gene with an lfsr of 0.119 and a P-value of 5.49 × 10−6 in the atrial appendage of the heart (Table 2; Figs. 1 and 2). Increased predicted expression of PRMT7 associated with schizophrenia in 32 of 33 tissues in GTEx tested (Figs. 2 and 3). Effect sizes were also positive for PRMT7 associations with schizophrenia in 42 of 42 tissues tested in our S-PrediXcan application to the PGC data (Schizophrenia Working Group of the Psychiatric Genomics Consortium, 2014) (Fig. 3). Of the 42 tissues tested, 30 associations were statistically significant (P < 0.0012) after Bonferroni adjustment for the number of tissues tested and all 42 tissues had P < 0.05 (Table S1). We found no significant gene-tissue associations using the MESA or DLPFC models. While PRMT7 in atrial appendage had the lowest lfsr across all models, RP11-646C24.5 had a lower P-value (Fig. 1), but high lfsr in both pancreas (lfsr = 0.860) and sigmoid colon (lfsr = 0.851). Notably, the standard error in both of these tissues was over twice the size of that of PRMT7. Unlike more traditional false discovery rate approaches such as Bejamini–Hochberg, both effect size and standard error are used in an empirical Bayesian framework to calculate lfsr and thus the gene with the lowest P-value may not be the gene with the lowest lfsr (Stephens, 2017). We found no significant associations with the MultiXcan, cross-tissue model.

Table 2 Top PrediXcan results for schizophrenia in African Americans (GAIN cohort) by local false sign rate (lfsr).

PRMT7 makes up five of the top eight associated gene-tissue pairs by lfsr. BP are reported as transcription start site for the respective genes.

Gene	Beta	t	P	se(beta)	Tissue (predictor)	lfsr	CHR	BP	
PRMT7	0.10	4.56	5.49 × 10−6	0.022	Heart_Atrial_Appendage	0.119	16	68392457	
PRMT7	0.09	4.22	2.50 × 10−5	0.020	Cells_Transformed_fibroblasts	0.225	16	68392457	
PRMT7	0.08	4.00	6.63 × 10−5	0.019	Heart_Left_Ventricle	0.353	16	68392457	
PRMT7	0.11	4.23	2.47 × 10−5	0.025	Adrenal_Gland	0.365	16	68392457	
TBC1D2	0.12	4.24	2.34 × 10−5	0.028	Brain_Putamen_basal_ganglia	0.468	9	100961311	
NPC1	0.07	3.83	1.29 × 10−4	0.019	Thyroid	0.484	18	21086148	
EIF2S2P3	0.08	3.81	1.41 × 10−4	0.022	Brain_Amygdala	0.558	10	94428502	
PRMT7	0.08	3.73	1.97 × 10−4	0.020	Cells_EBV-transformed_lymphocytes	0.578	16	68392457	

Figure 1 PrediXcan association results for schizophrenia in GAIN African Americans.

Each point on the Manhattan (A) and Quantile-Quantile (B) plots represents one gene-tissue test for association with schizophrenia using GTEx version 7 gene expression prediction models. PRMT7 expression in atrial appendage of the heart is labeled in both plots since it had the lowest lfsr of all tissues (lfsr = 0.119). Predicted RP11-646C24.5 expression in pancreas and sigmoid colon associations are represented as the two points with lower P-values than PRMT7, respectively, but lfsr was greater than 0.8 for each association. Unlike more traditional false discovery rate approaches such as Bejamini–Hochberg, the gene with the lowest P-value may not be the gene with the lowest lfsr especially if the standard error of the effect size estimate is high (Stephens, 2017).

Figure 2 Predicted PRMT7 expression is higher in schizophrenia cases than controls in GAIN.

The violin plot represents the differences in density of predicted gene expression levels of PRMT7 between cases (SCZ) and controls in heart atrial appendage from GTEx (P = 5.49 × 10−6).

Figure 3 PRMT7 PrediXcan discovery (GAIN) and replication (PGC) results across tissue models.

In each bubble plot, the radius of the bubble is representative of the significance of PRMT7 association with SCZ. The color of the bubble represents the test statistic with blue representing a positive direction of effect and red representing a negative direction of effect.

Bipolar disorder gene-based association study

To develop a better understanding of the genetic mechanisms governing bipolar disorder in African Americans, we performed PrediXcan in a cohort of 1,019 individuals (671 controls and 348 Cases). Similar to our gene-based association study of schizophrenia, we performed our tests across the same 55 gene expression prediction models in our bipolar disorder study.

In the GAIN cohort of 1,019 African American individuals, no genes were identified to be significantly associated with bipolar disorder. Increased predicted expression of GREM2 in testis was the most associated (P = 2.20 × 10−5) gene-tissue pair with bipolar disorder (Fig. 4). KCNMB3 had the lowest lfsr at 0.919. We also found no significant associations with the MultiXcan, cross-tissue model.

Figure 4 PrediXcan association results for bipolar disorder in GAIN.

Each point on the Manhattan (A) and Quantile-Quantile (B) plots represents a gene-tissue association test for our study of bipolar disorder using GTEx models. GREM2 on chromosome 1 was the gene most associated with bipolar disorder in our study. All of the gene associations tests across 48 tissues in GTEx are plotted in (A) and (B). A total of 95% confidence intervals depicted by gold dotted lines (B).

Schizophrenia SNP-level association test

We performed a GWAS across greater than 12 million SNPs following imputation to help elucidate the role specific SNPs play in the genetics of schizophrenia in African Americans. We used the first 10 principal components as covariates for our logistic regression in order to adjust for population stratification in the cohort. In our SNP-level GWAS, we found no significantly associated SNPs; however, one of the most associated SNPs, rs8063446 (P = 2.66 × 10−6), is located at the PRMT7 locus (Fig. 5) While not genome-wide significant, the most associated SNP in our study was rs112845369 (P = 1.094 × 10−6) on chromosome 15.

Figure 5 LocusZoom plot of the PRMT7 locus in the GAIN GWAS for schizophrenia.

rs8063446 is found in SLC7A6OS and 514 bp upstream of PRMT7. In our PrediXcan analyses, we found that increased predicted expression of PRMT7 is associated with schizophrenia. rs8063446 is located in a linkage disequilibrium (LD) block with other SNPs associated with schizophrenia when plotted using 1000G AFR LD Population.

Bipolar disorder SNP-level association test

We also performed a logistic GWAS in over 12 million SNPs in an attempt to understand the role specific SNPs play in the genetics of bipolar disorder. We similarly used the first 10 principal components to adjust for population stratification in the bipolar disorder cohort. Similar to our findings in our schizophrenia GWAS, we identified no SNPs significantly associated with bipolar disorder after adjusting for multiple tests. rs10168049 on chromosome 2 was the most associated SNP (P = 1.04 × 10−6). While not significantly associated with bipolar disorder, rs10168049 has a MAF of 0.465 in African 1000G populations compared to those of European and East Asian populations with minor allele frequencies of 0.042 and 0.097, respectively (Fig. 6) (Sherry, 2001; The 1000 Genomes Project Consortium, 2015). All PrediXcan and GWAS summary statistics for both diseases are available at https://github.com/WheelerLab/Neuropsychiatric-Phenotypes.

Figure 6 rs10168049 frequency across 1000G populations.

Representative minor allele frequencies (MAFs) of rs10168049, which associated with bipolar disorder in GAIN (P = 1.04 × 10−6) in different populations from 1000G. The global MAF of this SNP in 1000G is 0.187; however, the MAF reaches up to 0.541 in the YRI (Yoruba people in Ibadan, Nigeria) population of 1000G. This figure was generated using the Geography of Genetic Variants Browser (Marcus & Novembre, 2017).

Discussion

We performed gene-level (PrediXcan) and SNP-level association studies for schizophrenia and bipolar disorder in African Americans from GAIN (Suarez et al., 2006; Manolio et al., 2007). We used summary statistics from the predominantly European PGC to replicate our findings (Schizophrenia Working Group of the Psychiatric Genomics Consortium, 2014; Psychiatric GWAS Consortium Bipolar Disorder Working Group, 2011).

A potential role for PRMT7 in schizophrenia

PRMT7 was significantly associated with schizophrenia in our PrediXcan analyses and one of the most associated SNPs in our GWAS is just upstream of the gene (Figs. 1 and 5). Schizophrenia was associated with increased expression of PRMT7 in 32 of 33 tissues in which it was predicted in the GAIN cohort. When S-PrediXcan was applied to the PGC summary stats, increased expression of PRMT7 was associated with schizophrenia in all 42 tissues in which expression was predicted (Fig. 3). PRMT7 made up five of the eight most associated gene-tissue pairs (Table 2). PRMT7 has previously been associated with SBIDDS syndrome, an intellectual disability syndrome (Agolini et al., 2018). Its function in the disorder remains unclear, but PRMT7 has a functional role in neuronal differentiation, which could be a potential mechanism to explore further (Dhar et al., 2012).

While not found to be significantly associated with schizophrenia in brain tissues, the association of PRMT7 in adrenal gland and other vascular system organs highlights the sharing of eQTLs across tissues. Recently, genes in both colon and adrenal gland were identified to be significantly associated with schizophrenia (Gamazon et al., 2019). Gamazon et al. highlight the opportunity to better understand the genetic mechanisms of neuropsychiatric diseases outside of the context of the central nervous system. A larger sample size would be needed to elucidate expression correlations and potential co-expression networks underlying African American neuropsychiatric traits.

Growing need for diversity in GWAS and genetic prediction models

In our GWAS of bipolar disorder, rs10168049 was the most significantly associated SNP. This SNP has not been implicated in previous studies, and its higher MAF in African populations compared to European and Asian populations demonstrates how GWAS in African ancestry populations may tag key loci missed in European-only studies (Fig. 6). In addition to not being identified in any published GWAS, rs10168049 is not present in any of the 55 prediction models we used to impute gene expression levels (MacArthur et al., 2017). As a result, this SNP did not contribute to predicted expression levels in our gene-based association tests. The association of this SNP with bipolar disorder needs to be replicated in larger studies; however, the lack of associations in other GWAS in the GWAS Catalog (MacArthur et al., 2017) may also be a result of ascertainment bias due to the field’s focus on European populations (Lachance & Tishkoff, 2014).

Version 7 of the GTEx predictors we used contain data exclusively from individuals of European descent. These models are not optimal for predicting expression in African American cohorts (Mogil et al., 2018; Mikhaylova & Thornton, 2019). While they offer power driven by sample size, they do not include models with African ancestry-specific alleles that might affect susceptibility to neuropsychiatric traits in this population.

The use of prediction models from monocytes in MESA offers advantages with respect to similar ancestry, but at the loss of nearly half of the sample size of many GTEx tissues. To ideally predict expression in African American cohorts, prediction models built in more tissues from African ancestry reference transcriptomes are needed. Moreover, future ancestry-specific models will not only increase accuracy of expression prediction, but they will also create opportunities for different methods, such as local ancestry mapping, to be applied to expression prediction by accounting for recent admixture within African American cohorts (Zhong, Perera & Gamazon, 2019).

Conclusion

Information from this study provides promising insight into the genetic architecture of gene expression underlying two neuropsychiatric disorders in African Americans. The results of our study were curbed by the small sample size of GAIN. With just over 2,100 individuals in our study of schizophrenia and 1,000 individuals in our study bipolar disorder, our findings were limited in power compared to larger European studies nearly two orders of magnitude greater in size (Schizophrenia Working Group of the Psychiatric Genomics Consortium, 2014; Stahl et al., 2019).

The size and diversity of our prediction models further hindered our ability to identify novel genes associated with these disorders. The MESA models, the most diverse of our predictors, were still limited in tissue type (monocytes only) and size at 233 African American individuals, 352 Hispanic individuals, and 578 European individuals (Mogil et al., 2018). The GTEx and CommonMind predictors we used, while generated from a larger sample size than the MESA African American cohort for many tissues, were made from exclusively European individuals (GTEx Consortium, 2017; Huckins et al., 2019). These key limitations highlight the need to increase the number of individuals from diverse populations in the study of neuropsychiatric genomics. To best characterize the molecular mechanisms that govern complex traits in diverse populations, diverse models, reference panels, and study subjects need to be included in genomics research.

Supplemental Information

Supplemental Information 1 Principal component analysis of schizophrenia genotype data.

We performed principal component analysis on the GAIN cohort merged with three populations from version three of the HapMap Project. Each point on the plot represents one individual in the study plotted across axes for their first and second principal components. The three HapMap populations plotted are Chinese in Beijing and Japanese in Tokyo (ASN), European ancestry in Utah (CEU), and Yoruba people in Ibadan, Nigeria (YRI).

Click here for additional data file.

Supplemental Information 2 Principal component analysis of bipolar disorder genotype data.

We performed principal component analysis on the GAIN cohort merged with three populations from version three of the HapMap Project. Each point on the plot represents one individual in the study plotted across axes for their first and second principal components. The three HapMap populations plotted are Chinese in Beijing and Japanese in Tokyo (ASN), European ancestry in Utah (CEU), and Yoruba people in Ibadan, Nigeria (YRI).

Click here for additional data file.

Supplemental Information 3 Comparison of minor allele frequencies (MAFs) across imputation reference panels.

We imputed genotypes using the University of Michigan Imputation Server using either 1000G or CAAPA as the reference panel. (A-D) depict the MAFs of SNPs from the GAIN schizophrenia study. We saw a similar pattern of MAFs in the GAIN data of the bipolar disorder study. (A) depicts the MAF of SNPs at the intersection of each reference panel before filtering by r2 > 0.8 and MAF > 0.01. (B) Depicts MAFs of SNPs in 1000G and CAAPA from (A) that passed the filters of r2 > 0.8 and MAF > 0.01and were included in the GTEx prediction models across 44 tissues. (C) shows a plot of the MAFs of filtered SNPs from 1000G and CAAPA found in the MESA predictors. (D) shows a plot of the MAFs of filtered SNPs from 1000G and CAAPA that were included in our GWAS.

Click here for additional data file.

Supplemental Information 4 S-PrediXcan results of PGC schizophrenia data for PRMT7.

The table includes the results for PRMT7 in our S-PrediXcan application to the PGC GWAS summary statistics across 42 tissues in which gene expression was predicted.

Click here for additional data file.

Additional Information and Declarations

Competing Interests

Author Contributions

Human Ethics

Data Availability

The authors declare that they have no competing interests.

Peter N. Fiorica conceived and designed the experiments, performed the experiments, analyzed the data, contributed reagents/materials/analysis tools, prepared figures and/or tables, authored or reviewed drafts of the paper, approved the final draft.

Heather E. Wheeler conceived and designed the experiments, analyzed the data, contributed reagents/materials/analysis tools, authored or reviewed drafts of the paper, approved the final draft.

The following information was supplied relating to ethical approvals (i.e., approving body and any reference numbers):

This project was approved by the Loyola University Chicago Institutional Review Board (project number 2014).

The following information was supplied regarding data availability:

All datasets analyzed are available from the NCBI dbGaP at https://www.ncbi.nlm.nih.gov/gap/ using the accession numbers phs000021.v3.p2, phs000167.v3.p1, and phs00017.v3.p1. Gene prediction models are available from PredictDB at http://predictdb.org/.

All scripts and summary statistics can be found at https://github.com/WheelerLab/Neuropsychiatric-Phenotypes/.

Funding support for the companion studies, Genome-Wide Association Study of Schizophrenia (GAIN) and Molecular Genetics of Schizophrenia—nonGAIN Sample (MGS_nonGAIN), was provided by Genomics Research Branch at NIMH see below) and the genotyping and analysis of samples was provided through the Genetic Association Information Network (GAIN) and under the MGS U01s: MH79469 and MH79470. Assistance with data cleaning was provided by the National Center for Biotechnology Information. The MGS dataset(s) used for the analyses described in this manuscript were obtained from the database of Genotype and Phenotype (dbGaP) found at https://www.ncbi.nlm.nih.gov/gap/ through dbGaP accession numbers phs000021.v2.p1 (GAIN) and phs000167.v1.p1 (nonGAIN). Samples and associated phenotype data for the MGS GWAS study were collected under the following grants: NIMH Schizophrenia Genetics Initiative U01s: MH46276 (CR Cloninger), MH46289 (C Kaufmann), and MH46318 (MT Tsuang); and MGS Part 1 (MGS1) and Part 2 (MGS2) R01s: MH67257 (NG Buccola), MH59588 (BJ Mowry), MH59571 (PV Gejman), MH59565 (Robert Freedman), MH59587 (F Amin), MH60870 (WF Byerley), MH59566 (DW Black), MH59586 (JM Silverman), MH61675 (DF Levinson), and MH60879 (CR Cloninger). Further details of collection sites, individuals, and institutions may be found in data supplemental Table 1 of Sanders et al. (2008; PMID: 18198266) and at the study dbGaP pages.

Funding support for the Whole Genome Association Study of Bipolar Disorder was provided by the National Institute of Mental Health (NIMH) and the genotyping of samples was provided through the Genetic Association Information Network (GAIN). The datasets used for the analyses described in this manuscript were obtained from the database of Genotypes and Phenotypes (dbGaP) found at http://www.ncbi.nlm.nih.gov/gap through dbGaP accession number phs000017.v3.p1. Samples and associated phenotype data for the Collaborative Genomic Study of Bipolar Disorder were provided by The NIMH Genetics Initiative for Bipolar Disorder. Data and biomaterials were collected in four projects that participated in NIMH Bipolar Disorder Genetics Initiative. From 1991 to 1998, the Principal Investigators and Co-Investigators were: Indiana University, Indianapolis, IN, U01 MH46282, John Nurnberger, M.D., Ph.D., Marvin Miller, M.D., and Elizabeth Bowman, M.D.; Washington University, St. Louis, MO, U01 MH46280, Theodore Reich, M.D., Allison Goate, Ph.D., and John Rice, Ph.D.; Johns Hopkins University, Baltimore, MD U01 MH46274, J. Raymond DePaulo, Jr, M.D., Sylvia Simpson, M.D., MPH, and Colin Stine, Ph.D.; NIMH Intramural Research Program, Clinical Neurogenetics Branch, Bethesda, MD, Elliot Gershon, M.D., Diane Kazuba, B.A., and Elizabeth Maxwell, M.S.W. Data and biomaterials were collected as part of 10 projects that participated in the NIMH Bipolar Disorder Genetics Initiative. From 1999 to 2003, the Principal Investigators and Co-Investigators were: Indiana University, Indianapolis, IN, R01 MH59545, John Nurnberger, M.D., Ph.D., Marvin J. Miller, M.D., Elizabeth S. Bowman, M.D., N. Leela Rau, M.D., P. Ryan Moe, M.D., Nalini Samavedy, M.D., Rif El-Mallakh, M.D. (at University of Louisville), Husseini Manji, M.D. (at Wayne State University), Debra A. Glitz, M.D. (at Wayne State University), Eric T. Meyer, M.S., Carrie Smiley, R.N., Tatiana Foroud, Ph.D., Leah Flury, M.S., Danielle M. Dick, Ph.D., Howard Edenberg, Ph.D.; Washington University, St. Louis, MO, R01 MH059534, John Rice, Ph.D, Theodore Reich, M.D., Allison Goate, Ph.D., Laura Bierut, M.D.; Johns Hopkins University, Baltimore, MD, R01 MH59533, Melvin McInnis M.D., J. Raymond DePaulo, Jr, M.D., Dean F. MacKinnon, M.D., Francis M. Mondimore, M.D., James B. Potash, M.D., Peter P. Zandi, Ph.D, Dimitrios Avramopoulos, and Jennifer Payne; University of Pennsylvania, PA, R01 MH59553, Wade Berrettini M.D., Ph.D.; University of California at Irvine, CA, R01 MH60068, William Byerley M.D., and Mark Vawter M.D.; University of Iowa, IA, R01 MH059548, William Coryell M.D., and Raymond Crowe M.D.; University of Chicago, IL, R01 MH59535, Elliot Gershon, M.D., Judith Badner Ph.D., Francis McMahon M.D., Chunyu Liu Ph.D., Alan Sanders M.D., Maria Caserta, Steven Dinwiddie M.D., Tu Nguyen, Donna Harakal; University of California at San Diego, CA, R01 MH59567, John Kelsoe, M.D., Rebecca McKinney, B.A.; Rush University, IL, R01 MH059556, William Scheftner M.D., Howard M. Kravitz, D.O., M.P.H., Diana Marta, B.S., Annette Vaughn-Brown, MSN, RN, and Laurie Bederow, MA; NIMH Intramural Research Program, Bethesda, MD, 1Z01MH002810-01, Francis J. McMahon, M.D., Layla Kassem, PsyD, Sevilla Detera-Wadleigh, Ph.D, Lisa Austin,Ph.D, Dennis L. Murphy, M.D.

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
