# Peer review of "Transcriptome association studies of neuropsychiatric traits in African Americans implicate PRMT7 in schizophrenia"

_PeerJ, doi:10.7717/peerj.7778_

## Round 0.1 · original submission · Major Revisions

The reviewers have appreciated the importance of this work as a step in addressing the lack of diversity in population genetic studies. They have, however, raised a few major concerns about the analysis, especially in appropriate and sufficient testing for multiple hypotheses and assessing false positive findings, which must be addressed before the paper can be further evaluated for publication.

Reviewer 1 ·

Basic reporting

no comment

Experimental design

The main issue in the experimental design is that the Bonferroni correction was calculated accounting for the number of models tested in each tissue, but the correct multiple testing correction was supposed to account for the number of models and the number of tissues tested. My suggestion is to use a false discovery rate approach to account for the correlation among the tests conducted.

It is nice to see that the authors investigated the concordance between GTEx v6 and v7. However, it is also confusing. GTEx v7 includes a larger number of individuals and it should be a more powerful dataset. My suggestion is to consider GTEx v7 as the primary dataset and include the comparison with GTEx v6 as a reliability check.

Additionally, there are two main limitations that the authors should at least acknowledge:
1) GTEx includes mainly individuals of European descent and the use of this dataset is surely not optimal to predict the transcriptomic profile of African-American subjects.
2) African-Americans are a recently-admixed population and their genetic background is a mosaic of African and European haplotypes that can be very different among individuals included in this ethnic group. Principal components are likely not enough to account for the local ancestry diversity among the African-American individuals investigated.

Validity of the findings

The authors reported seven genes. However, with the exception of ZNF562 (Brain Cerebellum-V6), the predictors identified are mostly related to tissues that are not likely to be involved in the pathogenesis of psychiatric disorders. This may be explained by the incorrect multiple testing correction. The authors should justify better the biological meaning of the tissues identified with respect to the disorders investigated.

It is also hard to follow whether the gene-tissue combinations identified actually replicate in the European-ancestry dataset. For example, Figure 3 does not show the direction of the associations.

An additional suggestion is to conduct a trans-ancestry meta-analysis between the African-American and European-ancestry datasets to uncover additional genes that may be shared between these ancestry groups.

Additional comments

Investigating diverse ancestry groups is a priority in human genetic research. The present manuscript surely contributes to this important goal. However, there are several major issues that the authors should address.

Reviewer 2 ·

Basic reporting

Overall, this paper is structured reasonably and written clearly. However, some places are a bit tricky to parse and methods could be expanded upon. An example of a tricky to parse place: the choice of reference panels for imputation is a major consideration for diverse populations, and the authors appear to attend to this, although the phrasing of the relevant sentences is not fully clear. Specifically, which reference panel was used for imputation? The relevant sentences conflict with each other on lines 106-109. I believe the authors mean 1000G was used for phasing followed by CAAPA for imputation? The latter would be a much more appropriate fit for imputation, as it is more ancestry-matched.

Experimental design

1. Could the authors clarify why they chose to include expression models based on tissues not from the brain for these two mental health phenotypes? It seems challenging to interpret why expression changes in other some of the other organs identified could be related to neuropsychiatric phenotypes. Restricting to only nervous system derived tissues would have decreased the multiple testing burden / reduced concern over irrelevant/false positives associations.

2. Expanding on my concern over false positives: the authors do not currently have a detailed discussion of explicitly how multiple testing was addressed. This is key, as many tests for the same phenotype were run across many tissues in multiple datasets, some of which the authors note had differing significance thresholds within them, which reads as though a 5% significance threshold allowed in each. I did not see specific discussion of controlling for false positives across the 4 datasets further. The additional concern for the potential of false positive associations due to the admixed nature of these subjects plus the ancestry mismatch between the expression data and individual data means this is especially key to control for.

Validity of the findings

I would appreciate a further discussion regarding the many associations obtained in the PrediXcan that did not replicate [paragraph starting on line 177]. I appreciate that this is an understudied population which may have some ancestry-specific hits, but 5 out of 7 hits not replicating with any published research seems striking. What is the authors’ explanation for this result and justification that these are not false positives?

Additional comments

This paper takes a stab at starting to address a significant issue that plagues the GWAS community: the lack of diverse populations in psychiatric genetics research. I commend the authors for contributing to rectifying the underrepresentation of association studies on African American populations with this work. While most of their analyses are underpowered due to limitations of samples of relevant ethnicities, the overall workflow (QC, choice of analyses) appears generally appropriate. I do have some questions about technical specifics that I would like addressed, however, before I can recommend wholehearted acceptance of this paper, as detailed here. Specifically, I am not fully convinced that the genes highlighted by the expression scans are all truly hits relevant to the phenotypes of interest (especially given their lack of replication in much larger datasets) and would like the authors to fully convince me that there is no concern for false positives. Overall, while few significant results emerge from this paper, it is a commendable early attempt to address the disparity in association studies conducted on African American individuals.

---

## Round 0.2 · Minor Revisions

The revised manuscript has addressed the major comments from the reviewers. There is, however, one minor issue raised regarding the presentation of the significant results, which should be addressed so the paper is fully ready for acceptance to PeerJ.

Reviewer 1 ·

Basic reporting

The authors adequately addressed my concerns.

Experimental design

The authors adequately addressed my concerns.

Validity of the findings

The authors adequately addressed my concerns.

Additional comments

The authors adequately addressed my concerns.

Reviewer 2 ·

Basic reporting

The text on the x-axis of Figure 3 is very small and thus hard to read.

Experimental design

I am satisfied with the additional analyses to address my and the other reviewer’s concerns over multiple testing correction.

Validity of the findings

I have one remaining comment with regard to the presentation of the most significant SNP, rs10168049 – the authors should clarify their language to make clear that though this is the lead SNP, it is not necessarily the causal SNP. For example, on lines 227-229 in the Discussion they say “…its low minor allele frequency in European populations suggests that it could have a larger functional impact in African populations”. The notable feature of this SNP is that the larger MAF in African-descent populations lends it increased power to be significant in GWAS, but does not necessarily suggest that it has a bigger functional impact. It has a better ability to be identified, and thus opportunity to tag whatever is the true causal variant.

Additional comments

As noted in the last round of review, I appreciate the authors’ early contribution to rectifying the underrepresentation of genetic studies on non-European populations with this work. These new results appear a lot more robust and easier to interpret with the imposing of more stringent false discovery thresholds and after the removal of superfluous data.

In sum, while few significant results emerge from the reanalysis, I have much more confidence in the gene that did survive the false discovery testing. For this reason, I now feel comfortable recommending this paper for publication in PeerJ.

---

## Round 0.3 · accepted · Accept

Thank you for the quick turnaround and addressing all the comments.